## Perspective

sexual violence; post-traumatic stress disorder (PTSD); complex trauma; mental health services; Democratic Republic of the Congo (DRC)

**Corresponding author:**
Achille Bapolisi;
Email: achilami@yahoo.fr

# A mental health emergency: A clinical and cultural response to sexual violence in the Democratic Republic of the Congo

Achille Bapolisi[1,2,3]

[1]Faculty of Medicine, Université Catholique de Bukavu, Bukavu, Democratic Republic of Congo; [2]Hôpital Provincial Général de Référence de Bukavu, Bukavu, Democratic Republic of Congo and [3]Adult Psychiatry Department and Institute of Neuroscience, Cliniques Universitaires Saint-Luc, Université Catholique de Louvain, Brussels, Belgium

## Abstract

For nearly three decades, the Democratic Republic of the Congo has endured armed conflict that has devastated its population, leaving a staggering number of survivors of sexual violence. This article draws on over a decade of clinical, academic and field experience to explore the psychosocial and public health challenges of caring for these survivors. Despite the high prevalence of post-traumatic disorders – often severe and complex – the mental health system remains gravely under-resourced. The article examines gaps in mental health services, highlights the clinical intricacies of trauma resulting from rape (including complex PTSD and dissociation) and critiques the uncritical export of Western therapeutic models to African contexts. Emphasizing the need for culturally grounded, integrative care, the author advocates for community-based, trauma-informed, inclusive and context-sensitive approaches that bridge clinical science and local healing traditions. This holistic vision is essential for restoring dignity and mental health to survivors and for building a resilient public health infrastructure in the DRC.

## Impact statement

Sexual violence in the Democratic Republic of the Congo (DRC) remains one of the most pressing humanitarian and public health challenges of our time. Beyond its devastating physical consequences, rape generates profound psychological and social trauma that often persists for decades. Despite this, mental health remains chronically underfunded and under-integrated within the national health system, with fewer than one mental health professional per 100,000 inhabitants. This article underscores the urgent need to strengthen psychosocial care for survivors of sexual violence in a context marked by conflict, poverty and fragile institutions.

The wider impact of this work is threefold. First, it highlights the critical gaps in mental health coverage, proposing a pyramidal model of care that empowers communities, integrates non-specialist providers and connects survivors to specialized services through structured referral pathways. This approach is not only feasible in resource-limited settings but also essential for reaching the majority of survivors who remain outside formal care systems.

Second, the article emphasizes the importance of cultural adaptation. Imported Western therapies, while evidence-based, often fail to resonate with local realities. By integrating traditional practices, community narratives and collective rituals with biomedical interventions, survivors can find care that is both clinically effective and culturally meaningful.

Finally, the article broadens the discourse on inclusivity and prevention. Survivors include not only women but also men, children, people with disabilities and LGBTQ+ individuals who face unique barriers to care. Moreover, sustainable solutions must address not only victims but also perpetrators, through preventive strategies, education and rehabilitation to break cycles of violence.

By combining clinical science, public health strategies and cultural resources, this work advocates for holistic, survivor-centered approaches that can inform global mental health practice in other conflict and post-conflict settings.

## Introduction

The Democratic Republic of the Congo (DRC) has been at war for nearly 30 years, resulting in, among other consequences, millions of documented and reported rapes across its territory (Baaz and Stern, 2013; Maedl, 2011). Tragically, these sexual assaults have led to a high prevalence of post-traumatic symptoms and complex clinical presentations(Johnson et al., 2010). For over a decade, I have been confronted with this complex issue in my capacity as a psychiatrist – clinically, where I have supported survivors of sexual violence(Bapolisi, 2017); in research, where I have investigated the vulnerability and resilience factors involved(Bapolisi et al., 2023; Bapolisi

et al., 2020); and through training and supervising many professionals engaged in survivor care. It is from this experience that I attempt to reflect on the multifaceted challenges of psychosocial care for survivors of sexual violence in the DRC's sociocultural context, while also considering the opportunities that could help address these challenges.

### Epidemiological considerations

Despite the media and humanitarian attention garnered by the many instances of sexual violence in the DRC over the past three decades, reliable epidemiological data remain scarce or poorly disseminated. In June 2013, the Congolese Ministry of Gender, Family and Children, with technical and financial support from the United Nations Population Fund (UNFPA), published a telling report citing 10,795 new cases of sexual violence for the year 2012 alone, spanning seven provinces (Bandundu, Bas Congo, Katanga, Kinshasa, North Kivu, Eastern Province and South Kivu) (Kighoma, 2013). Médecins Sans Frontières (MSF) reported that in 2020, 10,810 survivors of sexual violence received medical and/or psychological care in public health structures supported by the organization throughout the country (Médecin sans frontières, 2021).

All reports converge on a similar finding: the most conflict-affected areas also report the highest rates of sexual violence. However, regions seemingly spared from direct fighting also display surprisingly high figures. One of the most striking findings is the predominance of civilian perpetrators over military ones, a trend confirmed by recent data from Goma showing that most sexual assaults occur in community settings and are associated with significant physical and psychological consequences (Kiakuvue et al., 2024). Nevertheless, studies indicate that within armed groups, sexual violence is driven by complex social and psychological motives – including power assertion, group cohesion and the normalization of violence – revealing that wartime rape is not merely opportunistic but rooted in broader systems of domination and dehumanization (Bitenga Alexandre et al., 2021).

While these few reports underscore the scale of the problem in conflict-affected populations, they also highlight the fact that sexual violence is not confined to war zones nor solely perpetrated by armed men. Alarming as these numbers may be, they likely represent only the tip of the iceberg – far more cases remain hidden due to stigma, lack of access to care and insufficient awareness. Given the scarcity of available data and their geographical and temporal limitations, there is a pressing need to collect systematic data across the entire country and, if possible, through longitudinal studies, in order to better capture the magnitude of sexual violence and its mental health outcome.

### A public health perspective

From a public health standpoint, sexual violence exposes critical gaps in the integration of mental health into the broader health system. The first major gap is the severe shortage of both human and material resources. In the DRC, there are only 0.9 mental health professionals per 100,000 inhabitants, and less than 0.1% of the national health budget is allocated to mental health (Lora et al., 2020; World Health Organization, 2015).

A second gap is the low actual coverage of mental health care, which lags far behind already insufficient primary care services. A third and more subtle gap lies in the disconnect between survivors' real needs and their perception of these needs. Many survivors are unaware that psychological disorders exist or that support is even possible. This lack of knowledge is compounded by widespread stigma against mental illness and survivors of violence, as well as by material barriers such as costs and the absence of health insurance or subsidies.

To address these gaps, a participatory approach is essential: mental health "by all, for all and with all." This means involving all stakeholders – from community members to specialists – in both prevention and care. In our work with refugees, we employed a pyramidal model in which community services formed the base and specialists the top. In a study we conducted among refugees, we found that meeting basic needs such as housing, food and safety serves as a protective factor against mental disorders, whereas their absence is strongly associated with the onset of post-traumatic stress disorder (Bapolisi et al., 2020).

We advocate for the implementation of the Inter-Agency Standing Committee (IASC) Guidelines on Mental Health and Psychosocial Support in Emergency Settings (Inter-Agency Standing Committee, 2006), emphasizing the engagement of all stakeholders – from local communities to specialized mental health professionals. At the base of this pyramidal model, strengthening social support networks, conducting awareness-raising campaigns and mobilizing both financial and human resources dedicated to mental health are essential actions at the community level. Culturally rooted assets that enhance resilience should be actively leveraged. Clinically, non-specialist providers such as general practitioners and nurses should receive targeted training and supervision to manage mild-to-moderate cases, with clear referral pathways for more severe conditions requiring psychiatric care. With appropriate oversight, these providers can effectively utilize accessible therapeutic tools, including both pharmacological treatments and evidence-based psychotherapies (Figure 1).

While the proposed pyramidal and community-based model offers a comprehensive framework, its implementation in resource-limited and politically unstable contexts faces significant challenges. Structural barriers such as the shortage and high turnover of trained personnel, limited and irregular funding and weak institutional coordination often hinder the sustainability of interventions. Insecurity and population displacement further compromise continuity of care and supervision mechanisms.

To enhance feasibility, several strategies can be pursued. First, integrating mental health and psychosocial support (MHPSS) within existing primary healthcare and community structures can reduce costs and strengthen ownership. Task-shifting approaches – training community health workers and non-specialists to deliver basic interventions – have demonstrated effectiveness and adaptability in fragile contexts (Okoroafor and Christmals, 2023). Second, partnerships with local and international NGOs can help secure complementary resources, technical assistance and ongoing supervision. Third, advocacy at the governmental and donor levels is essential to ensure policy commitment and dedicated budget lines for mental health. Finally, community engagement through local leaders, survivors' groups and culturally grounded practices remains key to fostering trust, promoting service uptake and ensuring that interventions are contextually appropriate and sustainable despite political instability.

These recommendations are fully aligned with the *Programme National de Santé Mentale* and the *Stratégie Nationale de Lutte contre les Violences Basées sur le Genre* of the Democratic Republic of the Congo, providing an operational framework through which this pyramidal and community-based care model could be effectively integrated and scaled up nationwide.

The suggested pyramid model requires a functioning referral, collaboration and supervision network. Community workers

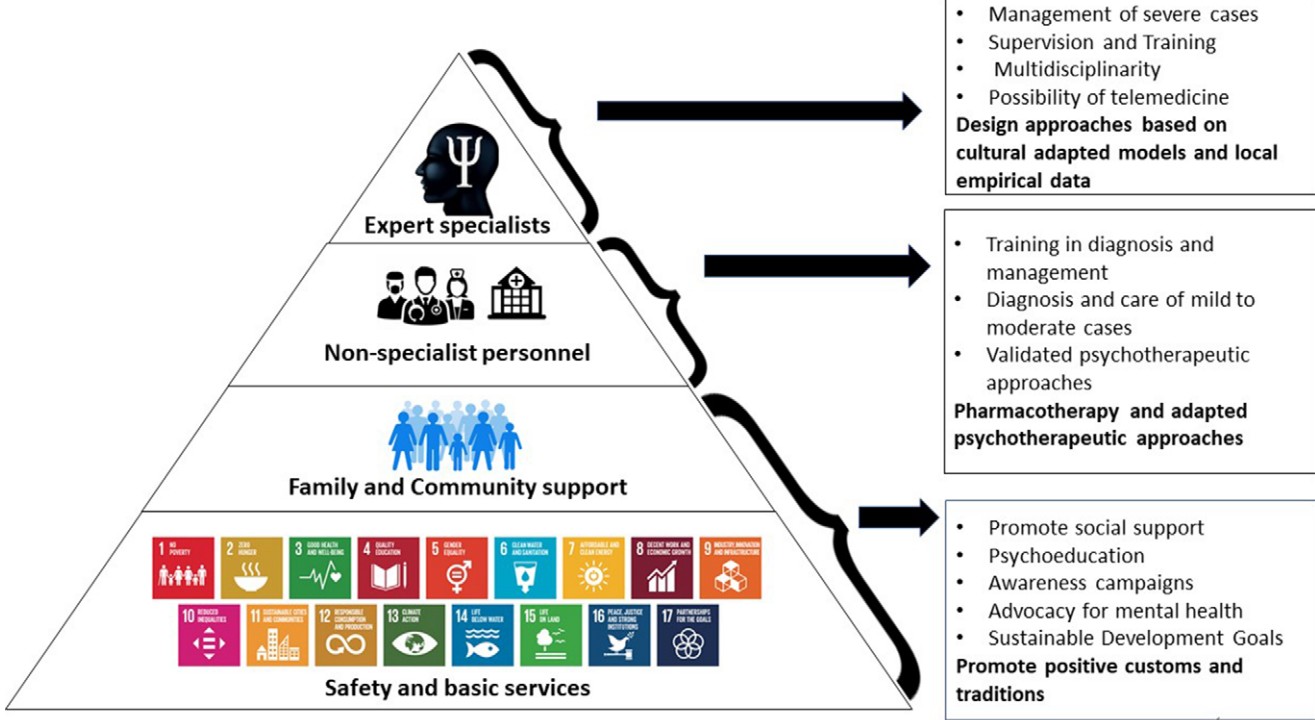

**Figure 1.** Pyramidal model of mental health and psychosocial support adapted for the Congolese context of sexual violence. The base includes safety and basic services addressing essential human needs and aligning with the Sustainable Development Goals (SDGs). The second level highlights family and community support through psychoeducation, social engagement and positive cultural practices. The intermediate layer involves non-specialist personnel (general practitioners, nurses, psychosocial assistants) providing training-based care for mild to moderate cases. At the top, specialized professionals manage complex and severe cases, ensure multidisciplinary supervision, integrate telemedicine and develop culturally adapted therapeutic models. The model promotes collaboration across all tiers, bridging community resilience and specialized mental health expertise in resource-limited and post-conflict settings.

trained in ethical standards and symptom recognition can drive awareness efforts. Their collaboration with local leaders – religious figures, traditional chiefs, teachers, police – is essential to encourage survivors to seek care. Once in health facilities, a multidisciplinary team (doctor, nurse, psychologist, psychosocial assistant) should carry out assessments, initial care and referrals when needed. An off-site consulting psychiatrist can provide ongoing training, supervision and support for complex cases (Figure 2).

While the proposed **task-shifting model** represents a pragmatic response to the scarcity of mental health specialists in the DRC, it is not without significant challenges. Sustaining the **quality and consistency of supervision** for non-specialist providers remains a major concern, particularly in remote or resource-limited areas. **Community workers and psychosocial assistants** are often exposed to distressing narratives, placing them at risk of **secondary or vicarious trauma** and eventual **burnout**. Furthermore, the establishment of **reliable referral pathways** is complicated by the fragmented nature of the health system and the limited availability of specialized services. Lay caregivers may also face **ethical and relational dilemmas**, as their proximity to survivors within their own communities can create tensions around confidentiality and social boundaries. Addressing these challenges requires ongoing **capacity-building, structured supervision, psychosocial support for caregivers** and the development of **clear ethical guidelines and referral mechanisms** to ensure both quality of care and provider well-being.

### Clinical challenges and perspectives

Sexual violence leads to highly debilitating psychopathological conditions, particularly post-traumatic stress disorder (PTSD).

Most training programs to date – including the mhGAP Humanitarian Intervention Guide (Organization, 2015) – focus on teaching providers to recognize PTSD and implement treatments such as pharmacotherapy (most commonly antidepressants) and psychotherapeutic techniques that can be taught to non-specialists, such as narrative exposure therapy (Robjant et al., 2019). While these approaches are generally useful in managing PTSD, their limitations become apparent in cases of PTSD stemming from sexual violence, due to the complexity of clinical presentation and care needs.

### Clinical complexity

PTSD is classically described in the DSM-5 as involving persistent symptoms following exposure to an event perceived as threatening to physical, moral integrity or life – namely, intrusive memories (flashbacks and nightmares), avoidance behaviors and neurovegetative hyperarousal (insomnia, exaggerated startle, hypervigilance) (American Psychiatric Association, 2013). The DSM-5 also includes non-specific features such as persistent negative emotions. Most treatment protocols and caregiver training on the psychopathological consequences of sexual violence are based on this symptomatology.

However, trauma resulting from sexual violence is often far more complex due to the particularly pathogenic and disintegrative nature of rape as a traumatic event. Beyond classical PTSD symptoms, severe trauma is frequently associated with comorbidities such as depression, anxiety disorders and addiction. More troubling still, sexual violence often leads to dissociative symptoms – depersonalization,

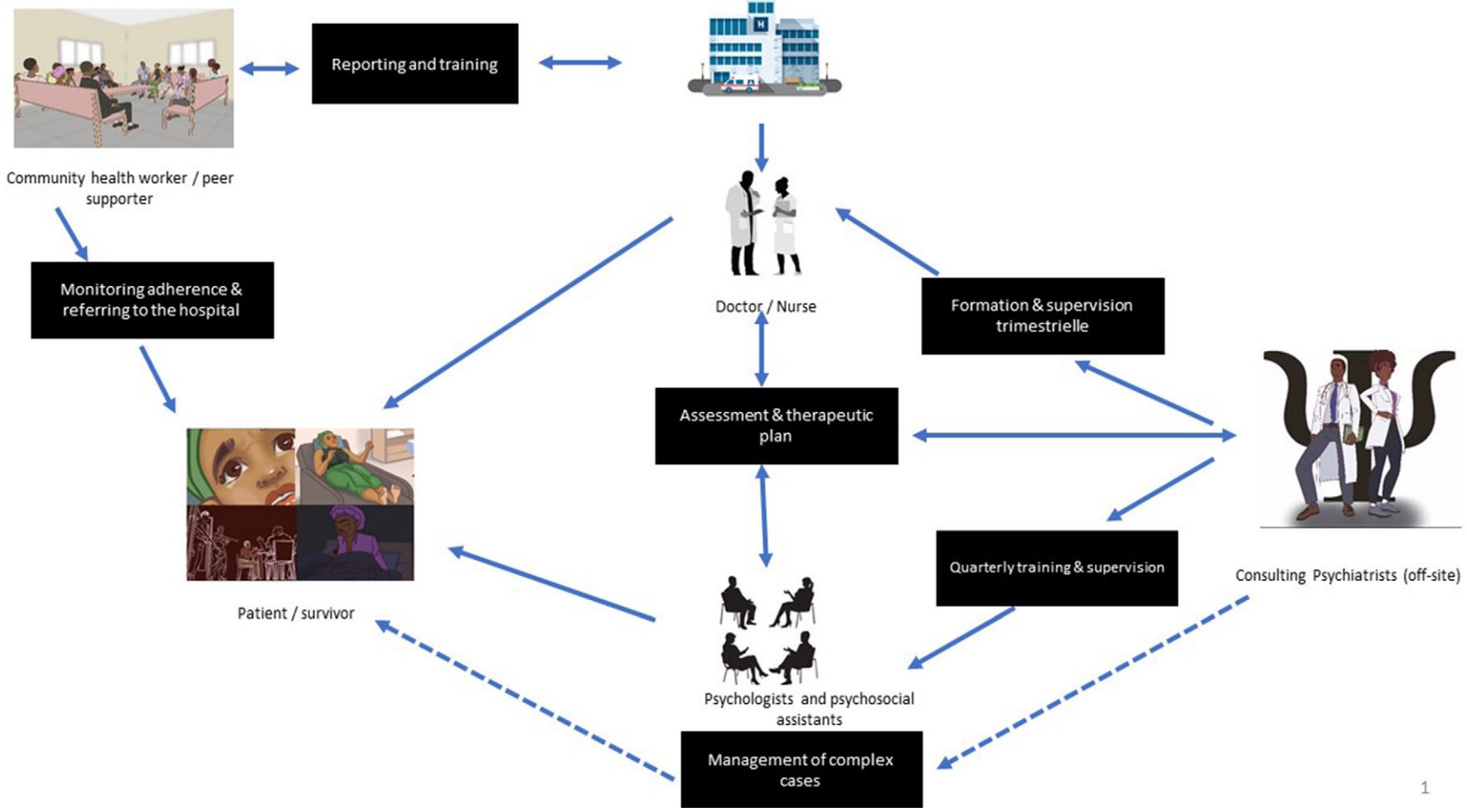

**Figure 2.** Proposed model for the organization of mental health and psychosocial support for survivors of sexual violence in the Democratic Republic of the Congo. The diagram depicts a collaborative and tiered network designed for the psychosocial management of survivors. At the community level, health workers and peer supporters ensure outreach, psychoeducation, adherence monitoring and referral of cases to health facilities. Within health structures, doctors and nurses perform clinical assessments and develop therapeutic plans in collaboration with psychologists and psychosocial assistants, who provide individualized and group interventions for complex cases. The system relies on continuous training, supervision and feedback mechanisms to strengthen capacity and maintain quality of care. Consulting psychiatrists (often off-site) provide supervision, training and guidance for frontline staff managing severe or complicated cases. This integrated model illustrates a community-based, multidisciplinary and contextually adapted approach that bridges community engagement, clinical expertise and sustainable mental health service delivery.

derealization, dissociative amnesia and, in some cases, dissociative identity disorder.

Clinicians have identified what is now termed Complex PTSD, especially in child survivors, which encompasses standard PTSD features along with a negative self-concept, emotional dysregulation and interpersonal disturbances (Roth et al., 1997). This clinical complexity is supported by neuroscience research using functional MRI, which identifies two distinct psychotrauma response patterns: emotional undermodulation and overmodulation (Yehuda et al., 2015). Emotional undermodulation involves hyperactivation of the amygdala and insula with decreased activity in the medial prefrontal cortex and rostral anterior cingulate cortex – resulting in classic PTSD symptoms such as fear, guilt, shame and anger. Overmodulation, on the other hand, features increased medial prefrontal activity and reduced amygdala/insula activity – linked to dissociative symptoms such as emotional numbing, derealization and depersonalization.

Significant advances have been made in psychopharmacology, favoring treatments with an optimal balance between therapeutic benefits and side effects. Recommended medications include selective serotonin reuptake inhibitors (SSRIs: fluoxetine, sertraline, paroxetine), serotonin–norepinephrine reuptake inhibitors (SNRIs: venlafaxine) and, in some guidelines, prazosin – an alpha-1 adrenergic antagonist used for PTSD-related nightmares. Amitriptyline (a tricyclic antidepressant) and mirtazapine (a noradrenergic and specific serotonergic antidepressant) have also shown efficacy (Williams et al., 2022).

Hence, clinicians must make informed therapeutic choices. Unfortunately, most of these medications are unavailable in the DRC, and when they are, physicians often lack the training to prescribe them properly. Training and supervision of general practitioners in prescribing these medications is essential, as is ensuring access through national health policy and financial support.

## Toward an integrated and contextualized psychotherapeutic approach

In a context where various psychotherapeutic schools claim primacy in understanding trauma, adopting a nuanced approach is urgent – especially regarding sexual violence in the DRC. While cognitive-behavioral approaches (CBT), including EMDR, narrative exposure and rumination-focused CBT, have strong evidence for PTSD (Grubaugh et al., 2021), their rigid application in complex trauma cases – frequent in rape – can be problematic without careful attention to cultural and psychological contexts.

The widespread use of standardized protocols by rapidly trained providers raises concerns: some techniques, such as explicit trauma narration, may be unsuitable for dissociative individuals or those from cultural settings where sexuality is deeply taboo. These well-intentioned practices may inadvertently retraumatize or reinforce avoidance.

A balanced therapeutic approach is crucial – one that fosters psychological safety through a strong therapeutic alliance, appropriate pacing and a stable framework, while also facilitating meaning-making and trauma integration. In many African contexts, personal identity and worldviews are deeply shaped by cultural narratives and collective representations. As such, effective treatment must be eclectic and culturally sensitive, grounded in core therapeutic principles: a trusting alliance, a secure and consistent structure, therapist credibility and the use of meaningful therapeutic rituals.

## Cultural considerations

Culture shapes both the experience of well-being and suffering. In the context of trauma – particularly sexual violence – ignoring culture is dangerous. Therapeutic approaches developed in the West often encounter legitimate limits in African settings. Western societies, valuing autonomy and introspection, favor analytical and intrapsychic frameworks. By contrast, traditional African societies prioritize the collective, symbolic language (myths, proverbs, stories), holistic cognition and metaphorical expressions of the psyche.

Tseng (2007) outlines six ways culture influences psychopathology: as a direct cause (pathogenic effect), a filter of defense mechanisms (pathoselective effect), a symptom modulator (pathoplastic effect), a generator of unique disorders (pathoelaborative effect), a frequency influencer (pathofacilitator effect) and a modulator of clinical response (pathoreactive effect). Thus, culture is central – not secondary – in treating sexual violence.

In the Democratic Republic of the Congo, responses to rape and mental illness often emerge at the crossroads of **traditional, religious and biomedical worldviews**. Traditional healers may interpret trauma through **spiritual or ancestral dimensions**, while religious leaders frame it within **moral or faith-based narratives**. The biomedical model, by contrast, explains psychological suffering through **psychological, neurobiological and social determinants**, emphasizing evidence-based interventions. These differing explanatory models may at times **generate tension**, especially when spiritual interpretations foster stigma or hinder access to care. Yet, each system holds deep **cultural and symbolic legitimacy** within its community. Addressing these tensions requires **cultural humility, dialog and collaboration** between healers, clergy and clinicians. Integrating culturally meaningful practices within an ethical, trauma-informed and evidence-based framework can strengthen trust, adherence and the overall effectiveness of mental health interventions for survivors of sexual violence in the Congolese context.

Recent experiences in the DRC and comparable contexts illustrate how evidence-based interventions can be meaningfully adapted to local cultural realities. For instance, Narrative Exposure Therapy has been modified to incorporate traditional storytelling and collective memory practices, making it more acceptable and effective among survivors of sexual violence (Robjant et al., 2019). Community-based counseling and psychoeducation sessions might rely on proverbs, songs and group discussions to normalize distress and promote resilience (Kohli et al., 2012). Rituals of purification and reconciliation, led by traditional or religious leaders, have been used to restore dignity and social reintegration for survivors, complementing biomedical care (Schneider, 2023). Faith-based support structures have also been integrated with psychotherapeutic approaches to reduce stigma and improve adherence (Pertek, 2024). Even pharmacological treatments, though limited in availability, gain acceptance when explained in culturally resonant terms by trained general practitioners. Finally, task-shifting models that train lay workers to provide psychosocial first aid and referrals have proven feasible in fragile systems, provided supervision and cultural competence are ensured (Connolly et al., 2021). These examples highlight that cultural adaptation is not an accessory to intervention design but a prerequisite for relevance, safety and sustainability of mental health care in the DRC.

An ethnopsychiatric approach is critical – it treats cultural specificities as therapeutic resources. For example, purification rituals may help restore a sense of dignity after rape. Clinicians must adopt a posture that respects both the patient's cultural world

and their individual narrative. This entails remaining true to one's therapeutic framework while adapting to the patient's worldview.

## Broadening the perspective: inclusive and culturally grounded prevention of sexual violence

Sexual violence in the Democratic Republic of the Congo remains both a public health and cultural challenge, affecting women, men and children across all regions. While women and girls are disproportionately affected, men, boys and other marginalized groups – including persons with disabilities, children and LGBTQ+ individuals – also endure profound trauma, often silenced by stigma, patriarchal norms and cultural taboos surrounding masculinity and sexuality. A comprehensive and equitable response must therefore integrate cultural understanding into prevention and care, ensuring inclusive, confidential and gender-sensitive services that respect local values while challenging harmful practices. Community engagement, traditional and religious leadership and culturally resonant awareness campaigns can help reshape collective attitudes toward dignity, gender equality and justice. At the same time, prevention must target the structural and cultural determinants of violence – inequality, impunity and the normalization of aggression – through education, dialog and the promotion of non-violent masculinities. Finally, the rehabilitation and reintegration of perpetrators, informed by psychosocial and restorative approaches, are essential to breaking the cycle of violence and fostering collective healing and resilience.

## Conclusion

Addressing the psychopathological consequences of sexual violence in the DRC is a massive challenge given the high prevalence and severity of clinical cases, the scarcity of material and human resources, the complexity of presentations and treatment approaches and the weight of cultural specificities. Meeting this challenge requires that care be embraced not only by mental health professionals but also by the general public, community leaders and non-specialist caregivers, in a network of collaboration, supervision and mutual support. Remarkable scientific advances in neuroscience, psychology and the humanities must be combined with local cultural elements to forge therapeutic approaches that are both effective and culturally coherent.

**Open peer review.** To view the open peer review materials for this article, please visit http://doi.org/10.1017/gmh.2025.10087.

**Data availability statement.** No new data were generated or analyzed in support of this research. Data sharing is therefore not applicable to this article.

**Author contribution.** AB conceived the study, conducted the literature review and wrote the initial draft of the manuscript. AB was also responsible for the integration of clinical, research and field experience, as well as for all subsequent revisions.

**Financial support.** This work received no specific grant from any funding agency, commercial or not-for-profit sectors. However, the author acknowledges institutional and logistical support from the Fonds National de Réparation des Victimes de Violences Sexuelles et des Crimes de Guerre (FONAREV), the Panzi Foundation and the Catholic University of Bukavu.

**Competing interests.** The author declares no conflicts of interest related to this manuscript.

**Ethics statement.** This article does not contain any studies with human participants or animals performed by the author. All clinical observations referenced derive from previously published studies or anonymized, aggregated data collected in accordance with international ethical guidelines and institutional approvals. IRB approval was not required.

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
