## [Reviewer Report]

I. General Assessment

This is a highly relevant and well-written perspective piece addressing a critical public health and mental health issue in the DRC. The author’s integration of clinical, research, and cultural insights is commendable, and the emphasis on culturally grounded approaches is both timely and necessary. The manuscript makes a valuable contribution to the literature by combining epidemiological, clinical, and cultural perspectives. However, certain aspects could be strengthened to enhance the clarity, evidence base, and applicability of the recommendations.

II. Major Comments

II.1. Epidemiological Data and Trends

While the manuscript presents key figures (e.g., UNFPA 2012, MSF 2020), more recent or longitudinal data—if available—would help readers understand whether the situation is improving, deteriorating, or stable.

It would also be useful to address methodological limitations in sexual violence prevalence studies in the DRC (underreporting, differences between conflict-related and non-conflict-related contexts).

II.2. Operational Feasibility of Proposed Models

The proposed pyramidal and community-based care model is compelling. However, the manuscript would benefit from a more detailed discussion on the feasibility of implementation in resource-limited and politically unstable settings, including potential barriers (e.g., workforce retention, funding, policy commitment) and strategies to overcome them.

II.3. Integration with Existing National Policies

The author should link recommendations to the Programme National de Santé Mentale and the Stratégie Nationale de Lutte contre les Violences Basées sur le Genre to show how these proposals could be integrated into existing frameworks.

II.4. Cultural Adaptation of Evidence-Based Interventions

While the manuscript critiques the direct transposition of Western therapeutic models, it could provide more concrete examples of culturally adapted interventions that have shown efficacy in the DRC or comparable contexts.

A comparative table summarizing intervention types, cultural adaptations, and evidence level would be helpful for practitioners.

II.5. Addressing Male Survivors and Other Vulnerable Groups

The focus is primarily on female survivors. Although they constitute the majority, male survivors and other marginalized groups (children, people with disabilities, LGBTQ+) also face significant stigma and access barriers. This dimension could be addressed to ensure inclusivity.

III. Minor Comments

III.1. Terminology Consistency: Ensure consistent use of terms such as “survivor” vs. “victim” throughout the text to maintain a survivor-centered approach.

III.2. I suggest you add a couple of recent, region-specific studies from the Kivu area to strengthen the epidemiological and contextual sections? Two relevant references you might include are:

Kiakuvue YN, Kanyere FS, Mukubu DM, Ruhindiza BM, Mukuku O. Sexual violence among female survivors in Goma, in the Democratic Republic of the Congo: epidemiology, clinical features, and circumstances of occurrence. Scientific Reports. 2024;14(1):14863.

Bitenga Alexandre A, Moke Mutondo K, Bazilashe Balegamire J, Emile A, Mukwege D. Motivations for sexual violence in armed conflicts: voices from combatants in eastern Democratic Republic of Congo. Med Confl Surviv. 2021;37(1):15–33.

These studies provide recent empirical data and local perspectives that would reinforce the paper’s claims about prevalence, perpetrator profiles, and contextual drivers of sexual violence in eastern DRC. You could cite them in the Epidemiological Considerations section (to complement the UNFPA/MSF figures) and in the Cultural / Motivations discussion where perpetrator profiles and local drivers are discussed.

III.3. Figures: Consider adding a visual representation of the proposed care model to improve reader understanding.

III.4. Language: A few sentences are lengthy and could be streamlined for clarity, particularly in the sections on neurobiological mechanisms of trauma.

IV. Conclusion

This manuscript addresses a pressing mental health and humanitarian issue in the DRC, offering a thoughtful and context-sensitive perspective. With minor revisions to strengthen the evidence base, expand inclusivity, and operationalize recommendations, it will make an important contribution to both academic and practical discourse on global mental health and sexual violence.

---

## [Reviewer Report]

Review of Manuscript GMH-2025-0227: “A mental health emergency: A Clinical and Cultural Response to Sexual Violence in the Democratic Republic of the Congo”

Drawing on over a decade of direct clinical, academic, and field experience, the author provides a cogent and compelling analysis of the profound mental health challenges facing survivors of sexual violence in the Democratic Republic of the Congo (DRC). The article effectively outlines the inadequacies of the current system, critiques the uncritical application of Western therapeutic models, and advocates for a more integrated, culturally grounded, and context-sensitive approach. The author offers a valuable perspective for clinicians, researchers, and policymakers working in conflict and post-conflict settings.

This article has a number of notable strengths, first and foremost is the author’s extensive, firsthand experience. This perspective is grounded in years of practice, which lends significant weight and credibility to the arguments presented. The discussion of the clinical complexities of trauma—particularly the distinction between classic PTSD and Complex PTSD, and the introduction of dissociative features and their neurobiological underpinnings—is important to underscore that specialized care is required. The author’s critique of exporting Western therapeutic models without cultural adaptation is also important. Evidence-based therapies are not always well-adapted to cultural contexts or for specific clinical presentations (e.g., dissociation). The author also offers an actionable framework for realistic solutions to this major global health challenge.

I would encourage the author to consider the following points:

The call to integrate local cultural elements and healing traditions is important, but under-developed. Beyond the mention of purification rituals, could the author provide further examples of the clinical framework being proposed? What are the challenges of such an approach. There should be at least a brief discussion on navigating potential tensions between traditional, religious, and biomedical approaches to rape, mental illness and its treatments.

The proposed task-shifting model is essential, but it is not without challenges. The author might briefly acknowledge the potential difficulties associated with this approach, such as the significant need for sustained, high-quality supervision for non-specialist providers, the risk of burnout in community workers exposed to severe trauma, and the complexities of establishing reliable referral pathways in a fractured health system. In addition, non-specialist exposure to secondary trauma is left unaddressed. Also not mentioned is the difficult role that lay caregivers are placed in when they are privy to sensitive information about members of their community.

Sexual violence disproportionately affects women and girls, and the author places his attention there, which is appropriate. However, the perspective could be broadened by explicitly mentioning the unique needs and barriers to care for male survivors, who are often overlooked in both research and clinical practice. In addition, there is no mention of preventive strategies to reduce the incidence of rape. What preventive measures could be adopted to assist with reducing the prevalence of rape? How are offenders treated and rehabilitated so they will not re-offend? Rape is something that involves not only the victim, but also the perpetrators, and while effective solutions must prioritize victims, there should also be attention paid to working to prevent people from becoming perpetrators.

This piece is both informative and moving. It speaks with authority and clarity on a critical issue in global mental health. Thank you for the opportunity to complete this review.

---

## [Reviewer Report]

Thank you for your edits and additions to this manuscript. All of my concerns have been addressed.